# Adoption of Total Neoadjuvant Therapy in the Treatment of Locally Advanced Rectal Cancer

**Madison L. Conces**  **and Amit Mahipal *** 

University Hospitals Seidman Cancer Center, Case Western Reserve University, Cleveland, OH 44106, USA
* Correspondence: amit.mahipal@uhhospitals.org; Tel.: +1-216-844-3951

**Abstract:** Local and metastatic recurrence are primary concerns following the treatment of locally advanced rectal cancer (LARC). Chemoradiation (CRT) can reduce the local recurrence rates and has subsequently moved to the neoadjuvant setting from the adjuvant setting. Pathological complete response (pCR) rates have also been noted to be greater in patients treated with neoadjuvant CRT prior to surgery. The standard approach to treating LARC would often involve CRT followed by surgery and optional adjuvant chemotherapy and remained the treatment paradigm for almost two decades. However, patients were often unable to complete adjuvant chemotherapy due to a decreased tolerance of chemotherapy following surgery, which led to upfront treatment with both CRT and chemotherapy, and total neoadjuvant therapy, or TNT, was created. The efficacy outcomes of local recurrence, disease-free survival, and pCR have improved in patients receiving TNT compared to the standard approach. Additionally, more recent data suggest a possible improvement in overall survival as well. Patients with a complete clinical response following TNT have the opportunity for watch-and-wait surveillance, allowing some patients to undergo organ preservation. Here, we discuss the clinical trials and studies that led to the adoption of TNT as the standard of care for LARC, with the possibility of watch-and-wait surveillance for patients achieving complete responses. We also review the possibility of overtreating some patients with LARC.

**Keywords:** rectal cancer; neoadjuvant therapy; chemotherapy

## 1. Introduction

Locally advanced rectal cancer (LARC) is defined as either stage II (clinical stage T3 or T4 and node-negative) or stage III (node-positive) rectal cancer. In 2023, there were approximately 800,000 new cases of rectal cancer diagnosed worldwide, with half staged as LARC [1]. The rates of diagnosis and mortality are decreasing overall for patients over the age of 65 due to cancer prevention and earlier diagnoses through screening. However, the rates of rectal cancer are increasing in patients under the age of 65, particularly under age 50 [2].

The goals for treating LARC center on preventing locoregional and metastatic recurrence while preserving quality of life for patients. Due to the location of the rectum deep in the pelvis, local recurrences are associated with high morbidity. Additionally, the tumor response to neoadjuvant treatment at surgery is critical to achieve negative margins. The presence of tumor cells within 2 mm of the circumferential resection margin (CRM) confers a poorer prognosis with higher rates of local recurrence, distant metastases, and overall survival (OS) [3].

While LARC was treated with the standard approach of neoadjuvant CRT, surgery, and adjuvant chemotherapy in the past, the transition to total neoadjuvant therapy (TNT) in recent years has shown promise in improving pCR, locoregional recurrence, metastatic recurrence, and even, possibly, overall survival [4–7].

## 2. Historical Approaches to LARC

### 2.1. Moving Radiation to the Neoadjuvant Setting

The concern for locoregional recurrence of LARC led to the adoption of radiation therapy (RT) as part of the treatment paradigm. Adjuvant RT was found to reduce locoregional recurrence (LRR) rates compared to chemotherapy alone, although no difference in disease-free survival (DFS) or OS was noted [8,9]. Subsequently, a study comparing neoadjuvant short-course radiotherapy (SCRT) and adjuvant RT found that patients treated with neoadjuvant SCRT had significantly lower local recurrence rates than those treated with adjuvant RT [10]. When neoadjuvant SCRT was compared to selective postoperative CRT in patients with involvement of the circumferential resection margin at surgery, patients who received neoadjuvant SCRT had, again, less local recurrence, with no difference in OS [11].

Two major trials, the Swedish Rectal Cancer Trial and the Dutch Study, investigated neoadjuvant radiation compared to surgery alone (Table 1). The Swedish Rectal Cancer Trial investigated neoadjuvant SCRT then surgery (anterior resection or abdominoperineal excision) compared to surgery alone. The local recurrence rate was almost three times higher in the surgery-alone cohort (26%) compared to neoadjuvant SCRT (9%, $p < 0.001$) at long-term follow-up [12]. The reduction in local recurrence was notable at all distances from the anal verge, although there was no statistically significant difference beyond 10 cm from the anal verge. Both the OS and cancer-specific survival were significantly greater in the SCRT group [12,13]. The Dutch Study also showed that treatment with SCRT prior to surgery (total mesorectal excision, TME) compared to surgery alone significantly decreased the rate of local recurrence from 11% to 5% ($p < 0.001$) [14–16]. Long-term follow-up analysis found that patients treated with neoadjuvant SCRT for stage III disease with negative circumferential resection margins at surgery had a significantly improved 10-year OS (50% vs. 40% treated with surgery alone, $p = 0.032$) [16].

**Table 1.** Selected clinical trials investigating neoadjuvant RT and CRT in rectal cancer.

| Trial | Therapy | Follow-up (Years, Median) | pCR, % | DFS, % | LRR, % | DM, % | OS, % | CSS, % |
|---|---|---|---|---|---|---|---|---|
| MRC CR07 and NCIC-CTG C016 [11] | Neoadjuvant RT | 5 | NA | 73.6 | 4.7 | 19 | 70.3 | NA |
| | Selective Adjuvant CRT | | | 66.7 | 11.5 | 21 | 67.9 | |
| Swedish Rectal Cancer Trial [12] | Neoadjuvant SCRT | 13 | NA | NA | 9 | 34 | 38 | 72 |
| | Surgery alone | | | | 26 | 34 | 30 | 62 |
| TME Dutch Study [16] | Neoadjuvant SCRT | 10 | NA | 74 | 5 | 25 | 48 | NA |
| | TME alone | | | 68 | 11 | 28 | 49 | |
| FFCD 9203 [17] | Neoadjuvant RT | 5 | 3.9 | 55.5 | 16.5 | NA | 67.9 | NA |
| | Neoadjuvant CRT | | 12.1 | 59.4 | 8.1 | | 67.4 | |
| CAO/ARO/AIO-94 (German Trial) [18] | Neoadjuvant CRT | 10 | 9 | 68.1 | 7.1 | 29.8 | 59.6 | NA |
| | Adjuvant CRT | | NA | 67.8 | 10.1 | 29.6 | 59.9 | |
| Akgun et al. [19] * | Neoadjuvant CRT | 5 | 13 | 75.2 | 7.4 | 13.6 | 79.8 | 87.5 |
| | Adjuvant CRT | | NA | 64.8 | 13.4 | 18.4 | 74.7 | 80 |
| NSABP-03 [20] | Neoadjuvant CRT | 5 | 15 | 64.7 | 10.7 | NA | 74.5 | NA |
| | Adjuvant CRT | | NA | 53.4 | 10.7 | | 65.6 | |

* Prospective, non-randomized study. pCR: pathological complete response, DFS: disease-free survival, OR: overall recurrence, LRR: locoregional recurrence, DM: distant metastasis rate, OS: overall survival, CSS: cancer-specific survival, MRC: Medical Research Council, NCIC-CTG: National Cancer Institute of Canada Clinical Trials Group, RT: radiotherapy, SCRT: short-course radiotherapy, TME: total mesorectal excision, FFCD: French-speaking Federation of Digestive Oncology, CRT: chemoradiation, NA: not available, NSABP: National Surgical Adjuvant Breast and Bowel Project.

### 2.2. Role of Neoadjuvant Chemoradiation

With the advent of neoadjuvant RT, the role of CRT in the preoperative setting for LARC also became established. A study investigating preoperative RT with or without

fluorouracil and leucovorin (5-FU/LV) also favored outcomes for the patients treated with CRT. While preoperative CRT had greater grade-3 and -4 acute toxicity, pCR was more frequent (11.4% vs. 3.6%, $p < 0.05$) and local recurrence rates were lower at 5 years (8.1% vs. 16.5%, $p < 0.05$). No difference was noted in either sphincter preservation or OS. Adjuvant chemotherapy with 5-FU/LV was given to all patients in the study, irrespective of neoadjuvant therapy [17].

The results of the German Rectal Cancer Study Group (CAO/ARO/AIO-94), originally published in 2004, became the standard combined modality treatment for LARC at the time. Patients with T3 or T4 node-positive disease were treated with either neoadjuvant long-course chemoradiation (CRT) (50.4 Gy with 5-FU 120 h infusions during the first and last weeks of radiation) followed by surgery and adjuvant chemotherapy or surgery followed by CRT and adjuvant chemotherapy. While there was no difference in OS, DFS, or distant metastases, the treatment toxicity (27% neoadjuvant CRT vs. 40% adjuvant CRT, $p = 0.001$) and local recurrence (7.1% neoadjuvant CRT vs. 10.1% adjuvant CRT, $p = 0.048$) were less in the neoadjuvant CRT cohort [18,21].

These results were similar to those of another non-randomized prospective study of neoadjuvant versus adjuvant CRT that found a decrease in LRR, cancer-specific survival, and late toxicity in the neoadjuvant CRT group. Additionally, CRT completion was greater in the neoadjuvant setting compared to the adjuvant setting [19].

### 2.3. Timing of Neoadjuvant Radiation in Relation to Surgery Matters

As neoadjuvant CRT has been adopted in treating LARC, we have gained a better understanding of how rectal tumors respond to radiation treatment. Most importantly, the tumor's response to radiation takes time. In a study by Habr-Gama et al. evaluating standard CRT (54 Gy with two cycles of 5-FU-based chemotherapy) to extended CRT (54 Gy with six cycles of 5-FU-based chemotherapy), tumors treated with extended CRT had greater rates of complete clinical and pathological responses. These tumors were also less likely to regain metabolic activity based on PET/CT imaging from 6 to 12 weeks after radiation completion, highlighting that tumors continue to respond to radiation for weeks following the last radiation treatment [22].

The Stockholm III Trial also found that patients who had a delay prior to surgery (after 4–8 weeks) after receiving SCRT had a lower tumor stage and greater pCR compared to patients who had immediate surgery (within 1 week) [23]. Additionally, the postoperative complications were significantly less in patients who had a delay prior to surgery [24].

### 2.4. Adjuvant Chemotherapy following Neoadjuvant Chemoradiation

While the National Comprehensive Cancer Network (NCCN) recommends adjuvant chemotherapy for patients who undergo CRT and surgery regardless of surgical pathology results, its role in improving outcomes is not well established [25]. The European Society for Medical Oncology (ESMO) Guidelines specifically do not routinely recommend adjuvant chemotherapy following neoadjuvant CRT and surgery [26]. The ADORE study, a phase-II trial, investigated the use of adjuvant infusional 5-FU/LV compared to FOLFOX following CRT and TME. There was no difference in OS at the 6-year follow-up, but adjuvant FOL-FOX did improve DFS [27]. The PROCTOR-SCRIPT study, a phase-III trial by the Dutch Colorectal Cancer Group, compared adjuvant chemotherapy with either infusional 5-FU or capecitabine to observation following CRT and TME. While no difference in OS, DFS, LRR, and distant recurrence was noted at the 5-year follow-up trial, the trial did close prematurely due to slow patient accrual [28].

A meta-analysis of patients treated with adjuvant chemotherapy following neoadjuvant CRT and surgery for LARC in European randomized, controlled, phase-III trials found that overall adjuvant 5-FU/LV did not improve OS, DFS, or distant recurrences compared to observation. However, a subgroup analysis did note that adjuvant chemotherapy improved DFS and distant recurrence for patients with tumors 10–15 cm from the anal verge compared to those with tumors < 10 cm from the anal verge [29]. Another

review analyzing data from rectal cancer trials of adjuvant chemotherapy did not find convincing evidence to support the use of adjuvant chemotherapy in patients who had already received neoadjuvant CRT [30].

*2.5. Role of Oxaliplatin as Concurrent Chemotherapy with Radiation*

All but one phase-III clinical trial showed increased toxicity with no clinical benefit for oxaliplatin as a concurrent chemotherapy in CRT. The STAR-01 study found that patients with mid-to-low rectal tumors treated with concomitant infusional 5-FU/LV and oxaliplatin during CRT had greater grade-3 and -4 toxicity (24% vs. 8%, $p < 0.001$) with no impact on primary tumor response compared to the patients who received only 5-FU/LV with CRT [31]. The ACCORD 12/0405 PRODIGE 2 study had a similar trial design, but capecitabine was given instead of infusional 5-FU/LV. No significant difference in local recurrence, OS, DFS, or grade-3 or -4 toxicity was noted between those who received CAPOX vs. capecitabine alone as concurrent chemotherapy during CRT [32]. The NSABP Trial R-04 treated patients with either infusional 5-FU/LV or capecitabine with or without oxaliplatin during CRT. Although patients treated with oxaliplatin had significantly more grade-3 or -4 toxicities, no additional benefit was seen in the pCR, LRR, DFS, OS, or surgical outcomes of either regimen with or without oxaliplatin [33,34].

The FOWARC Trial compared infusional 5-FU/LV plus RT, FOLFOX plus RT, and FOLFOX. In the long-term follow-up, no difference was seen in local recurrence rates, DFS, or OS in patients treated with FOLFOX with or without radiation compared to 5-FU/LV plus RT [35,36]. The JIAO (Liaoning Cancer Hospital) Trial also investigated adding oxaliplatin to capecitabine-based CRT. There was no difference in 3-year local recurrence, DFS, or OS between the groups. However, there was a significant decrease in distant metastatic rates at 3 years for patients who received oxaliplatin (16.5% vs. 28.16% treated with no oxaliplatin, $p = 0.045$) [37].

The PETACC-6 Trial investigated adding oxaliplatin to both neoadjuvant CRT with capecitabine and adjuvant chemotherapy with capecitabine. Patients who received oxaliplatin had higher rates of grade-3 and -4 adverse events, especially neuropathy and gastrointestinal toxicity during the neoadjuvant therapy. At 7 years, there was no difference in DFS or OS. It is worth noting that the completion of the protocol for the patients in the control arm (no oxaliplatin) was 68%, while the experimental arm with oxaliplatin was only completed by 54% of patients, suggesting the difficulty of receiving oxaliplatin during CRT and adjuvant chemotherapy [38].

The only phase-III trial to show an improvement in DFS with the addition of oxaliplatin was the CAO/ARO/AIO-04 study. This study investigated adding oxaliplatin to infusional 5-FU/LV in neoadjuvant CRT and adjuvant 5-FU/LV compared to CRT with infusional 5-FU/LV and adjuvant 5-FU/LV. At 3 years, adding oxaliplatin to 5-FU/LV-based neoadjuvant CRT and adjuvant chemotherapy improved the DFS (75.9% compared to 71.2% without oxaliplatin, $p = 0.03$) with similar rates of grade-3 and -4 adverse events [39]. A post hoc analysis evaluated the association of adherence to neoadjuvant CRT and adjuvant chemotherapy with DFS. A decrease in adherence to neoadjuvant CRT resulted in a decrease in DFS, but no difference in DFS was noted in patients treated with adjuvant chemotherapy [40].

## 3. Total Neoadjuvant Therapy: New Standard of Care for LARC

TNT has been adopted by many institutions as a standard of care for patients diagnosed with LARC based on multiple studies (Table 2). There are several different treatment sequences for TNT: SCRT then chemotherapy, chemotherapy then SCRT, CRT then chemotherapy, and chemotherapy then CRT. Additionally, TNT is now listed in the National Comprehensive Cancer Network (NCCN) Guidelines as treatment for LARC [25].

**Table 2.** Selected clinical trials investigating TNT in rectal cancer. Both STELLAR and UNICANCER-PRODIGE included adjuvant chemotherapy and are not TNT protocols.

| Trial | Therapy | Follow-Up (Years, Median) | pCR, % | DFS, % | LRR, % | LRFS, % | DM, % | DMFS, % | DrTF, % | OS, % | CSS, % |
|---|---|---|---|---|---|---|---|---|---|---|---|
| UNICANCER-PRODIGE 23 [5,6] * | CRT | 7 | 12 | 62.5 | 8.1 | NA | NA | 72 | NA | 76.1 | 79.6 |
| | INCT + CRT | | 28 | 67.6 | 5.3 | | | 79 | | 81.9 | 84.9 |
| RAPIDO [4,7] * | SCRT + CNCT | 5 | 28 | NA | 10 | NA | 23.0 | NA | 27.8 | 81.7 | NA |
| | CRT | | 14 | | 6 | | 30.4 | | 34.0 | 80.2 | |
| STELLAR [41] | SCRT + CNCT | 3 | 16.6 | 64.5 | 8.4 | NA | 22.9 | NA | NA | 86.5 | NA |
| | CRT | | 11.8 | 62.3 | 11.0 | | 24.7 | | | 75.1 | |
| Polish II [42] | SCRT + CNCT | 8 | 16 | 43 | 35 | NA | 36 | NA | NA | 49 | NA |
| | CRT | | 12 | 41 | 32 | | 34 | | | 49 | |
| OPRA [43] | INCT-CRT | 5 | 8 × | 72 | NA | 94 | NA | 82 | NA | 88 | NA |
| | CRT-CNCT | | 9 × | 71 | NA | 90 | NA | 79 | NA | 88 | NA |

* pCR data from 3-year follow-up. × Only patients without a complete clinical response underwent surgery; data from 3-year follow-up. pCR: pathological complete response, DFS: disease-free survival, LRR: locoregional recurrence, LRFS: local recurrence-free survival, DM: distant metastasis rate, DMFS: distant metastasis-free survival, DrTF: disease-related treatment failure, TME: total mesorectal excision, OS: overall survival, CSS: cancer-specific survival, SCRT: short-course radiation therapy, CRT: chemoradiation, CNCT: consolidation chemotherapy, UNICANCER-PRODIGE 23: Unicancer Gastrointestinal Group and Partenariat de Recherche en Oncologie Digestive, RAPIDO: rectal cancer and preoperative induction therapy followed by dedicated operation, STELLAR: short-term radiotherapy plus chemotherapy versus long-term chemoradiotherapy in locally advanced rectal cancer, not available, NA: POLISH II:, OPRA: organ preservation in patients with rectal adenocarcinoma, INCT: induction chemotherapy.

### 3.1. Addressing Locoregional Failure and Systemic Relapse in LARC

Total neoadjuvant therapy, or TNT, was developed to reduce locoregional failure (LRF) and systemic relapse in the treatment of rectal cancer (Figure 1). The UNICANCER-PRODIGE 23 was a phase-III study and one of the first trials comparing the historically standard approach of CRT followed by surgery and adjuvant chemotherapy to neoadjuvant chemotherapy and CRT. The experimental cohort in this study received induction chemotherapy with FOLFIRINOX, CRT, and then TME followed by six cycles of adjuvant FOLFOX [5]. When analyzing the most recent updated data at 7 years of follow-up, the PRODIGE-23 research group used the restricted median survival time (RMST) to analyze the primary endpoints. The median DFS was greater in the TNT cohort compared to the standard approach cohort (66.2 vs. 60.4 months, $p = 0.048$). The median metastasis-free survival (MFS) was also longer in the TNT cohort (69.3 vs. 62.1 months; $p = 0.011$) [6]. In addition, pCR rates at surgery more than doubled in the TNT cohort compared to the standard cohort (27.8% vs. 12.1%, $p < 0.001$) [5]. No difference was noted in local relapse rates. An overall survival benefit of 4.37 months for the TNT arm was noted using RMST ($p = 0.033$) [6]. UNICANCER-PRODIGE 23 showed that neoadjuvant chemotherapy with CRT rather than CRT alone resulted in improved outcomes. It is unclear how much the adjuvant FOLFOX given in the experimental cohort contributed to the differences in outcomes between the two cohorts, and this raises the question of whether the improvement in OS at 7 years could be entirely attributed to neoadjuvant chemotherapy and CRT. Additionally, the neoadjuvant chemotherapy was FOLIRINOX rather than the two-drug regimen of FOLFOX that is more commonly used in TNT.

The RAPIDO trial was another study that contributed to the adoption of TNT into practice. In this study, patients were randomized to either SCRT followed by CAPOX or the standard approach. At 3 years, the disease-related treatment failure (DRTF) was lower in the TNT arm (23.7%) compared to the standard approach (30.4%, $p = 0.019$) [4]. The pCR improved from 14.3% to 28.4% with TNT ($p < 0.001$). Distant metastases were also less, at 20% at 3 years for the TNT cohort compared to 26.8% ($p = 0.005$). While the LRF was similar at 3-years, the locoregional failure and LRR were greater in the TNT group compared to the standard approach group at the 5-year follow-up with LRF 12% vs. 8% ($p = 0.07$) and LRR

10% vs. 6% (*p* = 0.027), respectively [7]. The higher rate of LRR in the TNT group compared to those treated with TNT in other studies may be due to patients with higher-risk LARC being enrolled in this study (T4, N2, enlarged lateral nodes, extramural vascular invasion, mesorectal fascia involvement). While there was greater reduction in DRTF and distant metastases with TNT in this study, the results suggest that there may be limitations around SCRT maintaining locoregional control in patients with high-risk tumors.

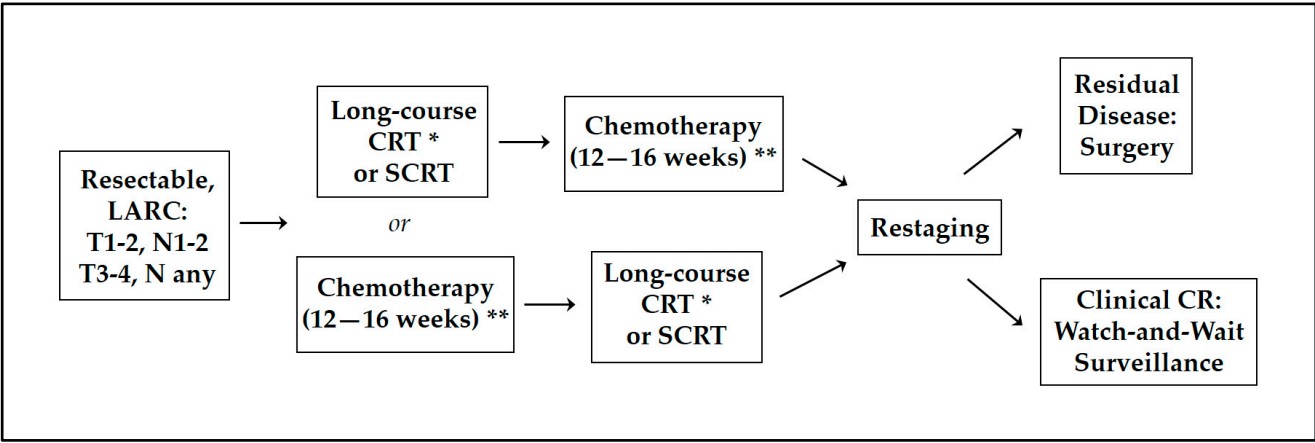

**Figure 1.** Treatment regimen for TNT. LARC: locally advanced rectal cancer, CRT: chemoradiation, SCRT: short-course radiotherapy, CR = complete response; * capecitabine or infusional 5-FU as concurrent chemotherapy; ** FOLFOX or CAPOX.

STELLAR, another phase-III study, aimed to determine whether SCRT followed by chemotherapy was non-inferior to CRT in the treatment of LARC with distal and middle rectal tumors. Patients in both arms received adjuvant chemotherapy. The 3-year DFS for the experimental arm and CRT was 64.5% and 62.5%, respectively (*p* < 0.001 for non-inferiority) [44]. No difference was found in MFS (77.1% experimental arm vs. 75.3% CRT, *p* = 0.475) or LRR (8.4% experimental arm vs. 11.0% CRT, *p* = 0.461). The 3-year OS was better in the experimental group (86.5% vs. 75.1%, *p* = 0.033). A subgroup analysis also found no difference in OS and progression-free survival (PFS) when clinicopathologic prognostic factors were considered. STELLAR found that SCRT followed by induction chemotherapy was an acceptable alternative approach to CRT in treating LARC. Additionally, the long-term follow-up from the Polish II study that evaluated CRT compared to SCRT followed by consolidation chemotherapy for T4 or fixed T3 rectal cancer found no difference in DFS, OS, local recurrence, or distant metastases at 8 years [41]. The study was not able to demonstrate that SCRT and consolidation chemotherapy were superior to CRT.

Results from an institutional database comparing patients who received induction chemotherapy and CRT versus those who only received neoadjuvant CRT found that at the 3-year follow-up, TNT had improved the DFS (83.5% vs. 71.4% CRT, *p* = 0.015), distant metastasis-free survival (DMFS, 84.3% vs. 75.2% CRT, *p* = 0.049), local recurrence-free survival (98.4% vs. 94.4% CRT, *p* = 0.048), and pCR rate (26.2% vs. 10% CRT, *p* < 0.001) [42]. Retrospective data have also shown that TNT results in greater complete responses in LARC [45,46].

While many TNT studies show improvement in outcomes, a phase-II randomized study (Grupo cancer de recto 3 study) of neoadjuvant CRT with concurrent CAPOX and adjuvant CAPOX compared to induction chemotherapy with CAPOX and CRT with CAPOX found no difference in pCR, complete resection rates, downstaging, or tumor regression [47]. The follow-up at 5 years also showed a similar DFS, OS, cumulative incidence of local recurrence, and distant metastases [48].

A study that performed an inter-trial comparison of the CAO/ARO/AIO-04 phase-III trial and CAO/ARO/AIO-12 phase-II trial did not find an improvement in DFS, cumulative incidence of LRR, cumulative incidence of distant metastasis, or OS [49].

### 3.2. Neoadjuvant Induction vs. Consolidation Chemotherapy

The order in which chemotherapy and CRT should be given during TNT is still under investigation. Studies thus far have suggested that consolidation chemotherapy may be a better approach as this gives the rectal tumor more time to respond to RT. The CAO/ARO/AIO-12 Trial, a phase-2 study comparing CRT with induction chemotherapy or consolidation chemotherapy, found a similar DFS (73% for both groups, $p = 0.82$), LRR (6% induction vs. 5% consolidation, $p = 0.67$), and incidence of distant metastases (18% induction vs. 16% induction, $p = 0.52$) at the 3-year follow-up. However, the pCR was greater in the consolidation chemotherapy group at 25% compared to 17% with induction chemotherapy ($p < 0.001$). Of note, both fluorouracil and oxaliplatin were given as concurrent chemotherapy with CRT in this trial [50,51]. Additionally, the results from OPRA trial demonstrated a similar DFS and OS regardless of sequence of CRT and chemotherapy, but the total mesorectal excision-free survival was higher with the CRT-first approach (54% vs. 39% with induction chemotherapy at the 5-year follow-up) [43,52].

### 3.3. Chemotherapy Completion Improves with TNT

While the debate on neoadjuvant induction versus consolidation chemotherapy is ongoing, we do know that treating patients with chemotherapy in the neoadjuvant setting increases the rate of completion and amount of chemotherapy received [53]. In a retrospective study comparing TNT and the standard approach, patients treated with TNT received a greater percentage of planned oxaliplatin and fluorouracil with fewer dose reductions than those in the adjuvant chemotherapy cohort [46]. The completion of CRT and adjuvant chemotherapy was also greater in the neoadjuvant CRT cohort compared to the adjuvant CRT cohort of the German Rectal Cancer Study [21]. In the Grupo cancer de recto 3 study, patients treated with neoadjuvant CAPOX and CRT received a greater amount of chemotherapy, with 96% starting induction chemotherapy and 93% completing it compared to those who received adjuvant chemotherapy, with only 71% starting adjuvant chemotherapy and 54% completing it [48].

## 4. Watch-and-Wait Method

Patients with a complete clinical response to neoadjuvant therapy based on imaging and clinical examination may have the opportunity to undergo watch-and-wait surveillance instead of proceeding to surgery (Figure 1).

### 4.1. Morbidity of Surgery

The morbidity of rectal cancer surgery can be significant. Depending on the type of surgery indicated, patients may require a colostomy with a permanent colostomy required for those undergoing abdominoperineal resection (APR). Sexual dysfunction is very common in patients following rectal cancer surgery, with some studies showing estimations as high as 90% [54–58]. Urinary dysfunction is also common. One study evaluating patients 5 years following TME found that 38% still experienced urinary dysfunction [59]. Low anterior resection syndrome (LARS) is an outcome specific to rectal cancer surgery where patients are burdened with unpredictable bowel movements such as urgency, loss of bowel control, and pain. Approximately 50% or more of patients still experience LARS over 13 years from their rectal cancer surgery [60,61]. Anastomotic leaks occur in approximately 10–30% of patients, with higher rates in APR over TME, and the post-operative death rates are 2–5% [62–64].

### 4.2. Organ Preservation

Watch-and-wait (WW) surveillance for organ preservation, pioneered by Dr. Habr-Gama, offers a non-surgical approach to patients whose tumors have had a complete clinical response (CR) to neoadjuvant therapy. An initial study of patients with resectable distal rectal adenocarcinoma were treated with neoadjuvant CRT. Those who had a complete CR underwent observation, with 100% OS and 92% DFS at 5 years compared to 88% OS

and 83% DFS for those who had an incomplete CR requiring radical surgical resection [65]. A follow-up study found that approximately half of the patients had a complete CR, with 31% having a recurrence. Only 2 of the 28 patients that had a recurrence had disease unamenable to surgery. Salvage therapy was possible in ≥90% of recurrences [66].

The International Watch-and-Wait Database (IWWD) has also provided long-term follow-up data on patients undergoing watch-and-wait surveillance. One study using this database noted that 88% of local regrowth occurred in the first 2 years, with 97% of local regrowth occurring in the bowel wall, emphasizing the importance of endoscopic surveillance in these patients. Local unsalvageable disease was rare [67]. Another study found that patients with no disease recurrence at 2 years continued to have a low probability of local or metastatic recurrence at 3 to 5 years on WW surveillance, suggesting that the intensity of surveillance could be reduced during that time [68]. Data from the IWWD also show that patients with complete CR on first reassessment after treatment completion compared to those with complete CR on later reassessment have no difference in 2- or 5-year organ preservation, DMFS rates, or 2-year OS [69].

## 5. Organ Preservation following TNT

The OPRA trial investigated WW surveillance following TNT in LARC. No difference was found in DFS, DMFS, or OS between patients treated in the experimental arm with TNT followed by WW and those treated with the standard approach [52]. Organ preservation with WW at 5 years was 54% in patients treated with CRT then consolidation chemotherapy, compared to 39% in those treated with induction chemotherapy then CRT ($p = 0.012$), with a 5-year DFS of 71% and 72%, respectively. This study showed that non-operative management in complete responders can be an option. At 5 years, survival after TME was the same at 64% for patients who initially required TME following restaging and those who underwent TME following regrowth, emphasizing again that salvage therapy is possible in patients who experience regrowth while on WW surveillance [43].

If organ preservation is a priority, then CRT followed by consolidation chemotherapy may be the best TNT sequence to ensure longer response times of rectal tumors to CRT [43,51,52].

## 6. TNT Is Not for Everyone

While TNT has a crucial role in improving outcomes for patients with LARC, this approach is not for every patient and can lead to overtreatment in some patients with LARC.

### 6.1. Mismatch-Repair Deficiency

There are promising results in a phase-2 study into single-agent dostarlimab, an anti-PD-1 monoclonal antibody, treating mismatch repair-deficient LARC. All 12 patients treated in the study had a complete CR with at least 6 months of follow-up [70]. While longer follow-up is needed, the results thus far are promising, especially since mismatch-repair deficient rectal tumors have been shown to be less sensitive to chemotherapy [71–73]. Despite its low incidence in rectal cancer, testing for mismatch-repair deficiency should now be routinely performed in all LARC cases [74].

### 6.2. Morbidity of Radiation

Radiation can also cause significant morbidity for patients undergoing treatment for LARC. Long-term bowel function was noted to be worse in patients in the Swedish Rectal Cancer Trial who underwent high-dose RT with increased bowel frequency, incontinence of stool, urgency, and emptying difficulties [75]. Additionally, 30% of patients treated with high-dose RT reported an impaired social life due to bowel dysfunction compared to 10% in the surgery-alone group in this study. Small bowel obstructions were also more common in patients who had undergone RT [76]. Bowel obstructions have also been noted to be significantly greater in patients receiving adjuvant RT compared to neoadjuvant RT [10].

The late side-effects of neoadjuvant SCRT compared to TME included significantly higher rates of fecal incontinence, pad wearing because of incontinence, anal blood loss,

and mucus loss in the patients who had undergone radiation compared to the TME-alone group. This study also noted that satisfaction with bowel function was significantly lower among patients treated with SCRT, with a greater impact of bowel dysfunction on daily activities [77]. Another study by Marijnen et al. found that patients treated with neoadjuvant SCRT had more sexual dysfunction, a slower recovery of bowel function, and impaired daily activity postoperatively compared to the TME-alone group [78]. A negative long-term impact on anorectal function has also been documented in several other studies investigating adverse effects at long-term follow-up [79,80]. Urinary incontinence is also a long-term adverse effect from RT for LARC, with one study also suggesting that high-dose RT may lead to increased cardiovascular morbidity [79,80].

An analysis of second cancer occurrences in patients enrolled on the Uppsala trial (completed 1985) and the Swedish Rectal Cancer Trial (completed 1990) found that, while there was an increased risk of second cancer occurrence in patients treated with RT for rectal cancer noted primarily in organs within or adjacent to irradiated tissue, the effect of radiation reducing the risk of local recurrence outweighed the risk of second cancer occurrence [81].

Radiation therapy also increases the risk of pelvic fracture. Baxter et al. noted a 5-year increased pelvic fracture risk in older women undergoing radiation therapy (11.2%) compared to those who did not undergo RT (8.7%) [82]. These risks should be discussed with patients, since pelvic fractures are a cause of significant morbidity and mortality in people over the age of 50, particularly women [83–85]. Other risks of pelvic RT include hematologic toxicity from radiation to the pelvic bone marrow, as well as infertility and premature menopause, issues particularly important as the rates of early onset rectal cancer are on the rise [86–92].

### 6.3. Intermediate-Risk LARC

The PROSPECT trial highlights the importance of catering treatment for LARC to the patient. The trial showed that neoadjuvant chemotherapy with the selective use of CRT was noninferior to the standard approach in patients with clinically staged T2 node-positive, T3 node-negative, or T3 node-positive rectal cancer at 5-year follow-up. The patients included in this study had only intermediate risk tumors (clinical stage T2 node-positive, T3 both node-positive and -negative, indication for CRT, tumor amenable to sphincter-sparing surgery). Those with clinical T4 tumors were excluded, as were patients with multiple enlarged lymph nodes, tumors requiring APR, and distal tumors. The experimental and standard cohorts at 5 years had a DFS of 80.8% vs. 78.6%, a local recurrence rate of 98.2% vs. 98.4%, and an OS of 89.5% vs. 90.2%, respectively. There was also no difference in complete (R0) rectal resection, low anterior resection rate, pCR, and positive radial margin between the two groups. Of the patients in the study randomized to FOLFOX with selective pelvic CRT, 91% of patients did not require CRT. Grade ≥ 3 adverse events were greater in the neoadjuvant setting for those treated with FOLFOX first, while CRT had greater grade ≥ 3 adverse events during adjuvant treatment. Both long-term sexual function and short-term bowel function were better in the neoadjuvant FOLFOX group, although only about 40% of the patients in the study completed the quality-of-life evaluation [93]. The results of PROSPECT suggest that there are patients with intermediate-risk LARC who are being overtreated with TNT. It is also worth noting that many of these patients would not meet the criteria for neoadjuvant radiotherapy according to ESMO Guidelines [26].

The OPERA study investigated organ preservation following the intensification of radiotherapy with either a boost of external beam radiotherapy at 9 Gy in five fractions or a boost with contact X-ray brachytherapy (90 Gy in three fractions) in patients with LARC who had already received CRT. The rectal cancers included were cT2, cT3a, T3b, <5 cm in diameter, cN0 or cN1 < 8 mm. At 38 months of median follow-up, the 3-year organ preservation rate was 59% in the external beam radiotherapy group compared to 81% in the X-ray brachytherapy group (HR 0.36, $p$ = 0.0026). The difference in organ preservation between the two groups was pronounced for rectal tumors < 3 cm, with organ preservation

rates for external beam radiotherapy and X-ray brachytherapy of 63% and 97%, respectively (HR 0.07, *p* = 0.012). No significant difference was seen in organ preservation between the two groups for tumors ≥ 3 cm [94]. This study demonstrates another potential treatment option in obtaining organ preservation without TNT for some patients with LARC.

A study of patients with LARC from the National Cancer Database from 2004 to 2015 compared those treated with TNT or neoadjuvant CRT followed by surgery with or without adjuvant chemotherapy. All patients had clinical stage T3 or T4 or node-positive disease. No significant difference in OS, pCR, or negative circumferential resection margin (CRM) was found between the two groups [95]. While this was a retrospective study, the results do still suggest that not all patients receive clinical benefits from TNT over the standard approach of neoadjuvant CRT and adjuvant chemotherapy.

## 7. Role of Additional Agents in TNT

A few other approaches to TNT have been investigated by adding either bevacizumab or cetuximab. The TRUST trial was a phase-II study investigating TNT with bevacizumab in LARC. Patients were treated with FOLFOXIRI and bevacizumab as induction chemotherapy, followed by CRT with concurrent capecitabine and bevacizumab, and then surgery 8 weeks after CRT completion. The pCR rate was 36.4%, and the DFS at 2 years was 80.45%. However, the capecitabine during the CRT schedule had to be modified due to 23.1% of patients having grade-3 palmar-plantar erythrodysesthesia and proctitis [96].

Another phase-II study, the EXPERT-C Trial, evaluated the addition of cetuximab to TNT. Patients with LARC received induction chemotherapy with CAPOX, then neoadjuvant RT with capecitabine, TME, and adjuvant CAPOX. Patients were randomly assigned 1:1 to receive weekly cetuximab during neoadjuvant CAPOX and CRT or the control treatment. Both skin toxicity and diarrhea were more frequent in the cetuximab cohort. While there was no difference in complete response or PFS in patients with KRAS wild-type LARC, significant improvements were noted in radiologic response after chemotherapy (51% no cetuximab vs. 71% cetuximab, *p* = 0.038) and CRT (75% no cetuximab vs. 93% cetuximab, *p* = 0.028), as well as in OS (HR 0.27, *p* = 0.034) [97]. Unfortunately, this improvement in OS for patients with KRAS wild-type was lost at the 5-year follow-up [98].

## 8. Rise in Early Onset Rectal Cancer

The discussion around treating patients with TNT and the potential for organ preservation is especially important for patients diagnosed with early onset rectal cancer (EORC). The cases of early onset (<50 years) colorectal cancer have been on the rise, with an estimated 2% annual increase [2,99,100]. The rates of EORC are rising faster than those of colon cancer, with one study predicting that 23% of rectal cancers will be in early onset patients by the year 2023, compared to 11% of colon cancers [101].

The diagnosis and treatment of EORC have a significant impact on the quality of life for this patient population. The support needed includes medical as well as social and economic support, since many are in early adulthood with fewer resources and social networks established.

## 9. Conclusions and Future Directions

Our ability to effectively treat LARC has improved with the introduction of TNT. Patients whose tumors have a clinical CR may have the option of WW surveillance instead of surgery, with the potential of significantly improving their long-term quality of life. As we learn more about which tumors respond best to different treatment modalities in TNT, our treatment paradigm will need to be adjusted. For example, SCRT may not be the best option for high-risk, bulky rectal tumors. Additionally, neoadjuvant chemotherapy, rather than TNT, for patients with intermediate-risk LARC may be adequate therapy. While they are beyond the scope of this review article, there are also studies looking into improved radiation therapy techniques for better local control and lower toxicity, such as the implementation of volumetric modulated arc therapy with a simultaneous

integrated boost [102]. Studies investigating the use of circulating tumor deoxyribonucleic acid (ctDNA) to help predict which patients would benefit from TNT or surgery versus WW surveillance will be helpful. There has been some research investigating the use of proteomics to predict responses to neoadjuvant CRT. One such study found a total of 915 proteins that had different expression profiles depending on whether the tumor had good or poor responses [103].

Ongoing studies that will hopefully contribute to our understanding of TNT outcomes include the KONCLUDE and JANUS trials. The KONCLUDE trial is a phase-III study comparing CRT followed by consolidation chemotherapy to the standard approach in patients with middle and lower rectal LARC (NCT02843191). The primary endpoints include pCR and 3-year DFS. While the design of KONCLUDE is similar to that of PRODIGE-23, FOLFOX is being used instead of FOLFIRINOX as neoadjuvant chemotherapy [104]. The JANUS trial is an ongoing phase-II study investigating clinical CR in patients with LARC treated with CRT and consolidation chemotherapy with either FOLFIRINOX or FOLFOX/CAPOX (NCT05610163) [105]. A prior phase-II study that evaluated only neoadjuvant XELOXIRI without CRT found acceptable local recurrence and DFS rates, but lower pCR than expected, at 7.7% [106].

Although we are awaiting the long-term follow-up results for dostarlimib for mismatch-repair deficient rectal tumors, the initial results are encouraging. Testing for mismatch-repair deficiency should routinely be performed for every LARC diagnosis, with immunotherapy or a clinical trial offered to those found to have mismatch-repair deficiency.

Developing the most effective treatment for LARC while balancing the impact on the quality of life for patients continues to be a focus of ongoing clinical trials as we work to further improve and individualize treatments for patients with LARC. The rising rates of EORC further emphasize the importance of taking such an approach, as many patients are facing different professional and personal obstacles while undergoing treatment for their cancer.

**Author Contributions:** Conceptualization: M.L.C. and A.M.; Investigation: M.L.C. and A.M.; Resources: M.L.C. and A.M.; Data curation: M.L.C. and A.M., Writing—original draft preparation: M.L.C.; Writing—review and editing: M.L.C. and A.M.; Supervision: A.M. All authors have read and agreed to the published version of the manuscript.

**Funding:** This research received no external funding.

**Conflicts of Interest:** The authors declare no conflicts of interest.

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
