# Peer review of "Adoption of Total Neoadjuvant Therapy in the Treatment of Locally Advanced Rectal Cancer"

_curroncol, doi:10.3390/curroncol31010024_

Round 1

Reviewer 1 Report

Comments and Suggestions for Authors

The topic is very important, up-to-date and in the scope of the Journal. The manuscript might be considered for publication after careful minor revisions:

1.     Please add a short section to address the occurrence/change of rectal cancer incidence in younger patient populations (under 40) and authors’ point of view of future directions

2.  Please add a short section to address the advantages of applying radiochemotherapeutic approaches using volumetric modulated arc therapy planning and simultaneous integrated boost.

3.     Please reinforce the manuscript/future directions reviewing most recent literature data on clinical, molecular and radiological/AI markers of response to chemoradiotherapy in rectal cancer similar to:

https://www.mdpi.com/1422-0067/24/20/15412

https://link.springer.com/article/10.1007/s12029-022-00909-w

Technical remarks:

1.     Please change the title of Tables 1 and 2 from:

Table 1. Select clinical trials investigating neoadjuvant RT and CRT in rectal cancer.

to

Table 1. Selected clinical trials investigating neoadjuvant RT and CRT in rectal cancer.

and

Table 2. Select clinical trials investigating TNT in rectal cancer. Both STELLAR and UNICANCERPRODIGE included adjuvant chemotherapy and are not TNT protocols

to

Table 2. Selected clinical trials investigating TNT in rectal cancer. Both STELLAR and UNICANCERPRODIGE included adjuvant chemotherapy and are not TNT protocols

 2.     Please delete the word “years” and “%” from the individual data in rows in Tables 1 and 2 as the units were defined in the column headings

Author Response

  1. Please add a short section to address the occurrence/change of rectal cancer incidence in younger patient populations (under 40) and authors’ point of view of future directions

Re: Thanks for the input. We have added the comment on young onset colorectal cancer in the discussion section.

  1. Please add a short section to address the advantages of applying radiochemotherapeutic approaches using volumetric modulated arc therapy planning and simultaneous integrated boost.

Re: While the whole topic is beyond the scope of this manuscript, we have added comment in the conclusion section.

  1. Please reinforce the manuscript/future directions reviewing most recent literature data on clinical, molecular and radiological/AI markers of response to chemoradiotherapy in rectal cancer similar to:

https://www.mdpi.com/1422-0067/24/20/15412

https://link.springer.com/article/10.1007/s12029-022-00909-w

Re: We have included the suggestion in the conclusion section.

Technical remarks:

  1. Please change the title of Tables 1 and 2 from:

Table 1. Select clinical trials investigating neoadjuvant RT and CRT in rectal cancer.

To Table 1. Selected clinical trials investigating neoadjuvant RT and CRT in rectal cancer.

and

Table 2. Select clinical trials investigating TNT in rectal cancer. Both STELLAR and UNICANCERPRODIGE included adjuvant chemotherapy and are not TNT protocols

To Table 2. Selected clinical trials investigating TNT in rectal cancer. Both STELLAR and UNICANCERPRODIGE included adjuvant chemotherapy and are not TNT protocols

Re: We have made the changes as suggested.

  1. Please delete the word “years” and “%” from the individual data in rows in Tables 1 and 2 as the units were defined in the column headings

Re: Changes have been made as per the reviewers excellent suggestions.

Reviewer 2 Report

Comments and Suggestions for Authors

Dear authors,

congratulations on this nice review.

I only have a few small comments from a European perspective, and I hope you can implement them:

"The standard approach in treating LARC involved CRT followed by surgery and adjuvant chemotherapy and remained the treatment paradigm for almost two decades"

-> I would write "optional adjuvant chemotherapy"

"However, additional treatment following surgery can be difficult for a proportion of patients"

-> I would write "adjuvant chemotherapy was associated with a lack of adherence"

"allowing some patients to forgo surgery altogether"

-> You may add the term "organ preservation" in the abstract

"When patients received adjuvant chemotherapy with or without RT, there was a decrease in locoregional recurrence rates for those who received RT compared to patients treated with chemotherapy alone without a differnce in DFS or OS."

-> Is that not exactly the same you wrote one sentence before? Please consider to delete this sentence.

"While adjuvant chemotherapy is recommended for patients who undergo CRT and  surgery, its role in improving outcomes is not well-established"

-> Is that true? The ESMO guideline (https://doi.org/10.1093/annonc/mdx224), in contrast to the NCCN guidelines, do not recommend adjuvant chemotherapy for all patients. Recommendation for adjuvant chemotherapy depents on pathological outcome, from our point of view. 

-> Please comment on this and add the reference to line 116/117 page 4

-> Please consider to add the analysis of Carvalho 10.1016/S1470-2045(17)30346-7 to you work (PMID: 28593861)

"The only phase III trial to show an improvement in DFS with the addition of oxaliplatin was the CAO/ARO/AIO-04 study"

-> You may add doi:10.1001/jamaoncol.2020.2394. This work tries to link the limited effect of oxaliplatin in the other trials to worse adherence to treatment and may provide an explanation why CAO-04 is the only positive trial.

Line 168

-> I would recommend to add here an explanation of the different treatment sequences, all being called TNT (SCRT+CTX or CTX+CRT or CRT+CTX) even if Figure 1 shows a graphically overview.

Line 217

-> Please add here the percentage of local failure.

Line 218

-> Please consider to change this sentence, as for me it may implicate that in the TNT "ARM" more advanced tumor were randomized compared to CRT what was not the case.

Line 248

-> https://doi.org/10.1016/j.radonc.2022.109455

-> TNT vs Oxaliplatin CRT showed better tumor regression but no survival benefit

-> post hoc analysis

Line 261

-> Please consider to add the impressive percentage number of organ preserveration of the OPRA trial already here.

Line 374

-> You may add in this paragraph that many patients in the PROSPECT trials are already no candidates for neoadjuvant radiotherapy, following the European guidelines, today.

Line 389

-> You may add that the long term quality of life data were only available from a small part of trial cohort, a potential bias.

Line 416

-> You may add a short paragraph adding the information that intensification of local radiotherapy is another possible to increase local recurrence and chance for organ preservation.

-> OPERA trial: https://doi.org/10.1016/S2468-1253(22)00392-2

Author Response

 1. "The standard approach in treating LARC involved CRT followed by surgery and adjuvant chemotherapy and remained the treatment paradigm for almost two decades"

-> I would write "optional adjuvant chemotherapy"

Re: We have edited the abstract as suggested.

2. "However, additional treatment following surgery can be difficult for a proportion of patients"

-> I would write "adjuvant chemotherapy was associated with a lack of adherence"

 Re: We have changed the wording of the abstract to better align with reviewers comments.

3. "allowing some patients to forgo surgery altogether"

-> You may add the term "organ preservation" in the abstract

 Re: The term has been added as suggested.

4. "When patients received adjuvant chemotherapy with or without RT, there was a decrease in locoregional recurrence rates for those who received RT compared to patients treated with chemotherapy alone without a differnce in DFS or OS."

-> Is that not exactly the same you wrote one sentence before? Please consider to delete this sentence.

 Re: Thanks for the suggestion, we have deleted this sentence.

5. "While adjuvant chemotherapy is recommended for patients who undergo CRT and  surgery, its role in improving outcomes is not well-established"

-> Is that true? The ESMO guideline (https://doi.org/10.1093/annonc/mdx224), in contrast to the NCCN guidelines, do not recommend adjuvant chemotherapy for all patients. Recommendation for adjuvant chemotherapy depents on pathological outcome, from our point of view.

-> Please comment on this and add the reference to line 116/117 page 4

Re: We have included the comment about ESMO guidelines. 

6. -> Please consider to add the analysis of Carvalho 10.1016/S1470-2045(17)30346-7 to you work (PMID: 28593861)

 Re: Has been added as suggested

7. "The only phase III trial to show an improvement in DFS with the addition of oxaliplatin was the CAO/ARO/AIO-04 study"

-> You may add doi:10.1001/jamaoncol.2020.2394. This work tries to link the limited effect of oxaliplatin in the other trials to worse adherence to treatment and may provide an explanation why CAO-04 is the only positive trial.

 Re: Thanks for the suggestion and we have added an sentence reflection this.

8. Line 168

-> I would recommend to add here an explanation of the different treatment sequences, all being called TNT (SCRT+CTX or CTX+CRT or CRT+CTX) even if Figure 1 shows a graphically overview.

 Re: We have edited the manuscript and figure 1 as suggested.

9. Line 217

-> Please add here the percentage of local failure.

 Re: Local failure rates are now included.

10. Line 218

-> Please consider to change this sentence, as for me it may implicate that in the TNT "ARM" more advanced tumor were randomized compared to CRT what was not the case.

 Re: We have modified the sentence to increase clarity.

11. Line 248

-> https://doi.org/10.1016/j.radonc.2022.109455

-> TNT vs Oxaliplatin CRT showed better tumor regression but no survival benefit

-> post hoc analysis

 Re: This information is now included in the manuscript.

12. Line 261 (Neoadjuvant Induction vs Consolidation Chemotherapy)

-> Please consider to add the impressive percentage number of organ preserveration of the OPRA trial already here.

 Re: We have now included organ preservation rates.

13. Line 374

-> You may add in this paragraph that many patients in the PROSPECT trials are already no candidates for neoadjuvant radiotherapy, following the European guidelines, today.

 Re: We have included this information per reviewer’s comments.

14. Line 389

-> You may add that the long term quality of life data were only available from a small part of trial cohort, a potential bias.

Re: Manuscript now includes the fact that quality of life data was only available for 40% of the patients.

15. Line 416

-> You may add a short paragraph adding the information that intensification of local radiotherapy is another possible to increase local recurrence and chance for organ preservation.

-> OPERA trial: https://doi.org/10.1016/S2468-1253(22)00392-2

 Re: We have included Opera trial as well as comment about intensification of local radiotherapy.

Reviewer 3 Report

Comments and Suggestions for Authors

Nice review, comprehensive description of the problem, history and future, based on trails' data. Some conclusions could be more supported by data from metanalyses.  

Author Response

Nice review, comprehensive description of the problem, history and future, based on trails' data. Some conclusions could be more supported by data from metanalyses.  

Re: We have modified the manuscript per the reviewer’s suggestions.

Round 2

Reviewer 1 Report

Comments and Suggestions for Authors

Thank you for the informed revisions.